# Transitions and challenges for people with Parkinson's and their family members: A qualitative study

Joy Read[1], Rachael Frost[2], Kate Walters[2], Remco Tuijt[2], Jill Manthorpe[3], Bev Maydon[4], Jennifer Pigott[1], Anette Schrag[1], Nathan Davies[2]*

1 Department of Clinical and Movement Neurosciences, UCL Queen Square Institute of Neurology, University College London, London, United Kingdom, 2 Research Department of Primary Care and Population Health, University College London, London, United Kingdom, 3 NIHR Policy Research Unit in Health and Social Care Workforce, King's College London, London, United Kingdom, 4 Member of Patient and Public Involvement (PPI Group), London, United Kingdom

* n.m.davies@ucl.ac.uk

## Abstract

### Objective

To explore the experiences and challenges of people with Parkinson's and their family members living in the community through the lens of their transitions to better understand the phases and changes in their lives.

### Design

Qualitative study using semi-structured interviews and analysed using codebook thematic analysis.

### Setting/participants

Purposive sampling was used in primary and secondary healthcare services across Southern England in 2019 to recruit 21 people with Parkinson's (aged between 45–89 years) and 17 family members (13 spouses and 4 adult children, aged between 26–79 years).

### Results

Participants' descriptions were classified in three main phases of transition from a place of health towards greater dependency on others: 1) 'Being told you are a person with Parkinson's' (early), 2) 'Living with Parkinson's' (mid), and 3) 'Increasing dependency' (decline). Seven sub-themes were identified to describe the transitions within these three phases: phase 1: receiving and accepting a diagnosis; navigating reactions; phase 2: changing social interactions and maintaining sense of self; information: wanting to know but not wanting to know; finding a place within the healthcare system; and 3: changes in roles and relationships; and increasingly dependent.

**Data Availability Statement:** All relevant data are within the manuscript and its Supporting Information files.

**Funding:** This report is independent research funded by the National Institute for Health and Care Research (NIHR) (Programme Grants for Applied Research, Personalised care for people with Parkinson's Disease: PD-Care, RP-PG-1016-20001). The views expressed in this publication are those of the authors and not necessarily those of the NHS, the National Institute for Health and care Research or the Department of Health.

**Competing interests:** The authors have declared that no competing interests exist.

## Conclusion

This study has identified points of change and means of supporting key transitions such as diagnosis, changes in social connections, and increased use of secondary healthcare services so that comprehensive, holistic, individualised and well-timed support can be put in place to maintain well-being.

## Introduction

Parkinson's is a progressive neurodegenerative condition more commonly occurring in people over the age of 60 with more men affected than women [1], and a large increase in prevalence and global burden has been reported [2]. Both motor and non-motor function are affected, impacting many physical, cognitive, emotional and psychosocial life domains [3, 4] consequently reducing quality of life [5]. It also significantly reduces carers' quality of life, with physical and emotional 'burden' reported among carers of those with greater disability [6, 7].

The concept of a transition is used in many studies of long-term health conditions to refer to a relocation within health and social care settings or to different services [8]. It is also used metaphorically to refer to the psychological adjustments to change or anticipated changes affecting an individual encountering a new or emerging health status, or to another person closely affected such as a family carer [9]. Both definitions are applicable to the lives of those with Parkinson's, but it is the latter that is the main concept used for the purpose of this paper.

A review of the nursing literature on transitions [10] suggested that transitions occur when life's circumstances change and require processes of adaptation and reorientation to integrate the changes into life. With a progressive condition such as Parkinson's, there is the need for constant adaptation and adjustment to changes in physical and cognitive function and ability [11, 12], as well as a loss of identity over time [13], for both the person with Parkinson's and those close to them, thus making the concept of transition potentially relevant. Transitions potentially include, for example, changes in employment status, increasingly complex pharmacological and non-pharmacological management, location of care, and overall ability and well-being in those with Parkinson's. This can mean continued experiences of limitation or losses of independence, altered self-efficacy, and changes in self-concept and self-esteem [14]. The disease trajectory in Parkinson's varies across individuals, with variation in both the symptoms that manifest and the timescale of symptom progression [15]. Symptoms of Parkinson's are also unpredictable and can fluctuate daily, with 'on'/'off' periods, where medications are no longer working optimally, and debilitating symptoms are no longer controlled, affecting many areas of life [16].

It is important to understand the lived experience of people with Parkinson's to enable those with Parkinson's, those closest to them, and those providing their care, to better anticipate and be prepared with effective and relevant strategies for optimal management as Parkinson's progresses. This study therefore aimed to explore the experiences of adults with Parkinson's living at home in the community, and the family members closest to them, which was analysed through the lens of transitions.

## Methods

### Design

An explorative qualitative study, using semi-structured interviews with people with Parkinson's and family members who supported them, allowing for an in-depth exploration of their lived experiences. Interviews were analysed using thematic analysis, with reporting guided by the Standards for Reporting Qualitative Research framework (COREQ) [17].

### Sampling and participants

Purposive sampling was used to ensure a range of ages, genders, ethnicities, marital status, home support, location, time since diagnosis, and level of disability as guided by the Schwab & England inventory [18]. For sampling we adopted the concept of information power [19], we were interested in interviewing a range of participants according to the above criteria. We continuously reflected on the quality and richness of the data collected. Towards the end of data collection we discussed as a team the depth of information we had received and agreed that it was appropriate to stop further recruitment of participants, confident we were able to answer our study aims and objectives. Participants were recruited through hospital outpatient clinics and General Practice (GP) surgeries (January —July 2019). Exclusion criteria were: Atypical Parkinsonism, living in a care home, lacked capacity to provide informed consent to an interview, or had a life expectancy of less than six months. There were no exclusion criteria for family members other than they had to be 18 years and over.

### Ethics

This study was given a favourable opinion by the London Queen's Square Research Ethics Committee (18/LO/1470). Written informed consent was obtained from all participants.

### Procedure

Recruitment of people with Parkinson's and their family members was through NHS Parkinson's services and primary care, with clinicians making initial approaches. Potential participants received an invitation letter, information leaflet and reply slip. Those indicating interest were screened, given the opportunity to ask questions, and an interview arranged with a member of the research team.

A topic guide for both people with Parkinson's and family members was developed, based on the study objectives and key literature [12, 20, 21] with input from members of the study's patient and public involvement (PPI) group, and refined during a pilot and the interview process. The open-ended interview questions explored participants' history and the challenges of having, or supporting someone with, Parkinson's, ways of managing these and what people thought might be helpful to better self-manage their condition. This included questions, for example, "Which are the most difficult aspects of your Parkinson's at the moment?"; "If you are finding your Parkinson's difficult what do you do?". Prompts and probes, including "Can you tell me more about that", or "How do you feel about. . ." were used to elicit more in-depth responses, and replies were summarised throughout enabling participants to ensure their responses were understood. The full topic guide is presented in S1 File.

After written informed consent had been obtained face-to-face interviews were carried out by either (anon) or (anon). The female researcher interviewers had healthcare and/or psychology backgrounds and were further trained for this study by a senior researcher experienced in

qualitative methods (anon), and therefore had the skills to build rapport and facilitate honest responses, of importance as researcher and participants were not known to one another. Participants were interviewed separately, except for two couple dyads. Apart from one couple, all took place in participants' homes. Interviews lasted up to 90 minutes (range 60–90 mins), and were audio-recorded, transcribed verbatim, de-identified, and all transcripts checked against source recording for accuracy.

### Analysis

Codebook thematic analysis, was applied to identify, analyse and report themes [22, 23]. Codebook thematic analysis was chosen as this was an explorative study, it allowed for an inductive approach to analysis as a team, helping us to construct common ideas and patterns within and across participants and present these as key themes. The analysis team was multidisciplinary spanning a range of clinical and academic expertise including neurology, primary care, gerontology, nursing, social care and psychology which allowed us to discuss a variety of ideas and perspectives about the data.

Transcripts were read and checked against the original audio recordings for accuracy, transcripts were then read repeatedly by (anon) and (anon) to build familiarity and an overview of the content. Five transcripts were coded by two authors and an initial coding list was developed. This coding list was discussed and agreed upon between (anon), (anon) and (anon). The code list was used to code all transcripts line-by-line using NVivo 12 by (anon) (anon) and (anon). Codes were reviewed and discussed throughout the analysis, revisiting previously coded transcripts to apply any new or check and refine codes. Although we did not originally plan (and therefore specifically ask about) the concept of transitions in interviews, from initial inductive analysis and team discussions it was identified as a conceptual framework that would organise relevant themes in a way that would have meaning from a theoretical and clinical perspective [9, 10]. Consequently, upon completion of coding, (anon), (anon) and (anon) met to discuss and develop initial ideas of themes and sub-themes. Following this all authors met on several occasions to discuss the interpretations including nuances and subtleties within the themes, and discuss sub-themes through the lens of transitions, resulting in the three main themes and seven subthemes presented below.

## Findings

The final sample (see Table 1) consisted of 21 people with Parkinson's (labelled 'P' in quotes), who ranged from 45 to 89 years, the majority [12] were male, living with female spouses, and had been diagnosed with Parkinson's for between 5 months and 20 years. Among the family members (labelled 'C' in quotes) 13 spouses and 4 adult children took part, ranging in age between 26 and 79 years. Participants were recruited from areas across Greater London and Hertfordshire, including rural areas.

### Themes

Participants with Parkinson's spoke of continuous changes and adaptations in response to different and progressive symptoms, from a place of health to a state of greater dependence, across three sequential, but overlapping, 'phases': phase 1) *Being told you are a person with Parkinson's (early)*, phase 2) *living with Parkinson's (mid)*, and phase 3) *increasing dependency (decline)*. These three phases and the sub-themes within these phases are represented in Fig 1 below.

**Being told you are a person with Parkinson's.** The first point of transition focussed on the time of diagnosis when participants were told they have a chronic neurodegenerative

**Table 1. Participants' demographic details and Parkinson's characteristics (n = 21).**

| Those with Parkinson's | | n = 21 |
|---|---|---|
| Gender | Men–n | 12 |
| | Women–n | 9 |
| Age | Mean–years | 72 |
| | Range–years | 45–89 |
| Marital status | Married | 17 |
| | Widowed | 4 |
| Ethnicity | White | 17 |
| | Indian | 3 |
| | Other Asian background | 1 |
| Living arrangements | With spouse | 14 |
| | With other family members | 2 |
| | Alone | 4 |
| | With live-in care worker | 1 |
| Duration since Parkinson's diagnosis | Mean–years duration | 8 years |
| | Range–years duration | 5 mo–20 yrs |
| Support of specialist nurse | Yes | 16 |
| | No | 4 |
| | Did not know | 1 |
| **Family members** | | n = 17 |
| Gender | Men–n | 7 |
| | Women–n | 10 |
| Age | Mean–years | 66 |
| | Range–years | 26–79 |
| Relationship to person with Parkinson's | Spouse | 13 |
| | Daughter/Son | 4 |
| Ethnicity | White | 15 |
| | Indian | 1 |
| | Pakistani | 1 |

disease; marking a period of becoming a person with Parkinson's. This was where presenting symptoms were managed, future implications considered, and the process of adjustment began. The sub-themes within this theme are *receiving and accepting a diagnosis;* and *navigating reactions*.

**Receiving and accepting a diagnosis.** Participants with a range of disease durations recalled receiving a diagnosis of Parkinson's as both as an internal process and as externally in the world involving noticing symptoms, going to the GP and subsequent referrals to specialists:

"*I was under the doctor for various things. And when I started my hand shaking, my common sense told me that it was the start of something. And I did research on it with the internet. And the word 'Parkinson's' kept coming up. [. . .] So I went to my GP and he diagnosed, he diagnosed–there was a letter where he [specialist] agrees with my doctor.*" (06P. Male, Diagnosed < 1 year)

Family members recalled greater perceived shock in the person with Parkinson's when the diagnosis was unexpected or sudden, for example when admitted to hospital for an unrelated

**Figure one:   Themes and subthemes**
**Health to dependence**

**Fig 1. Themes and subthemes.** The thick blue arrow reflects the movement though three phases as people transition through their Parkinson's. The smaller vertical arrows represent multiple acute episodes which fall along this trajectory and will impact someone's transitions. The faint line represents the undulating nature of Parkinson's.

matter or being told directly by a GP based on observations without investigations or specialist referral:

> "She took herself up the doctor's, and he said, 'hold your hands up'. [. . .] he said you've got Parkinson's, so that was the diagnosis. Next door here, there's [name], he's only 58 [. . .] he's just been diagnosed as well. But he had a proper test at the hospital, some sort of brain scan I believe." (14C. Son of female diagnosed 14 years)

Reassurance was gained from the chance to ask questions and undergo what were perceived as thorough investigations, and by empathetic experts:

> "He (doctor) was very warm. He was understanding. He had a lot of empathy with what [name] was going through. He's seeing it every day, but to us it was like he understood totally and was able to reassure us." (06C. Wife of male, diagnosed < 1 year)

For some younger participants or those who were at an earlier stage of family life with young children, without suspicions of having Parkinson's, reaching a diagnosis took longer, became more of a shock, and acceptance and adjustment were less straightforward:

> "It was sort of out of the blue, because I was initially diagnosed as having arthritis. And I was being treated for arthritis, but eventually they said I had Parkinson's, young onset Parkinson's. Well I thought I didn't, because I was quite healthy enough and everything, except

*for some joint pains. But slowly, gradually, the condition started sinking in.*" (01P. Male, Diagnosed 8 years)

**Navigating reactions.**  There were many accounts from those across ages, genders and disease durations of concerns around telling family members, friends and employers about their diagnosis. Others' ignorance was felt a potential risk due to being perceived as different, impaired, incompetent or unreliable:

*". . . nervous about telling others about it for what the other people might think, because there's generally a lot of ignorance about Parkinson's in society.*" (16P. Male, Diagnosed 3 years)

Participants had chosen to tell others about their diagnosis in various ways, and being open was seen as a positive way of helping people understand Parkinson's, particularly when symptoms began to affect normal activities:

"*I decided we would never tried [try] to conceal it. We decided to just be very upfront with it. And that worked, seems to have worked for us.*" (12P Male, Diagnosed 3 years)

However, others chose to keep their diagnosis hidden, either because of reluctance to accept the diagnosis or because of stigma:

"*And there is a stigma attached to the condition back home, where I come from (India). [. . .] So nobody would like, even my family wouldn't like to tell anybody else that I'm having this condition. So it's better off taking me as a drunk, rather than a PD patient.*" (01P male, Diagnosed 8 years)

There was particular reluctance to share the diagnosis when in paid employment. Fear of being misunderstood, with Parkinson's seen as an older person's condition, caused worry that ability and competence could be questioned. The potential risk of losing employment brought both short- and long-term implications for sense of identity and economic security:

"*My workplace doesn't know that I'm having this condition. [. . .] as long as I am working, as long as I can work, I'll keep on going until then, unless I really feel the need to tell them about it.*" (01P. Male, Diagnosed 8 years)

## Living with Parkinson's

The next transition was 'Living with Parkinson's' which included *changing social interactions and maintaining sense of self*, *information: wanting to know but not wanting to know*, and *finding a place in the healthcare system.*

**Changing social interactions and maintaining sense of self.**  Despite the strategies to manage others' reactions and to overcome growing social isolation, a gradual reduction in social activities and relationships was described, in part attributed to the impact of Parkinson's symptoms influencing how people felt about themselves in social situations.

Reduced mobility, risk of falls, regularly needing to use the toilet, and freezing where movement of a part of the body is impossible for a short time, made going out of the home more

problematic. Similarly, changes in memory, mood and speech affected interactions. Over-salivating, difficulties with dexterity or swallowing, for example, meant some were concerned with social dignity leading to greater self-consciousness both for the person with Parkinson's and those accompanying them:

> *"It's not easy to socialise in that situation and I'm very conscious as well. You know, when we go out, I've got to help her cut her food up and, yes, avoid spillages and things like that."* (10C. Husband of female, Diagnosed 2 years)

Additionally, fatigue, unpredictable symptoms, and the necessity of carefully timed medication to manage fluctuation of symptoms, and 'on'/'off' periods, made it both difficult to be spontaneous or to plan:

> *"It's very difficult to make an appointment in the morning at the moment because I know that at some point I'm likely to be shaky."* (17P Female, Diagnosed 15 years)

Despite the challenges of maintaining social contact and activities, the desire to feel 'normal', with a sense of purpose and satisfaction, and not defined by the condition, was demonstrated though the interactive hobbies and activities still undertaken, and sometimes adapted, such as exercise classes. For some, new social networks emerged from Parkinson's or carers' groups, but generally pre-existing friendships helped maintain a sense of connection. For others, faith communities were important and modifications helped maintain existing connections and attachments:

> *"I don't socially go out as much as what I used to. Well, I get visits from people. The band, being in the [church] band, they're a very caring band. And most recently, one Sunday morning, I was sitting here and I heard the band outside the window. The band had come to visit me."* (6P. 75 year old male, recently diagnosed)

**Information: Wanting to know, but not wanting to know.**   Many participants, across ages, genders and duration of Parkinson's, valued information about the condition and practicalities such as managing finances. However, views were mixed as how to best access information, whether directly from healthcare professionals as experts, or sourcing information themselves. Some felt some information provided at the time of diagnosis was deliberately restricted to protect the person with Parkinson's or was inadequate:

> *"I think if GPs could be more aware or if the Parkinson's nurse or, I don't know. I think if they could all just be aware that one might need just a little bit more help in understanding. Understanding what one can do and can't do and what the medication will do, what the side effects could be."* (12C. wife of male, Diagnosed 3 years)

In contrast, others found information from a range of sources was supportive and empowering:

> *"Our knowledge of Parkinson's was virtually nilch [nothing]. [. . .]. There's a lot of support around, lots of help and guidance, [. . .]. So, between us all, we can make a fair fist of it."* (13P. Male, Diagnosed 6 years)

Many successfully sought information themselves, through books, hospital leaflets, personal recommendations, charities, and online. The latter was often initiated and supported by extended family members, including grandchildren, which sometimes revealed a difference in desire for knowledge between family members:

*"Well, they're [children and grandchildren] always looking on, on the line, on this iPad, getting things to say. But it goes on. Frighten the life out of you. Read this, read that. [. . .] Since they knew I had Parkinson's, they all went up like an army. One's looking at this book, one's reading that one, one's. . ."* (15P. Male, Diagnosed 10 years)

Despite family members often seeking more information some acknowledged that explicit information could be upsetting to the person with Parkinson's:

*"Yes, information yes, but I don't suppose he'd want to talk about that, because it will only upset him, won't it?"* (04C. Son of male)

There were not only some different desires for information between the person with Parkinson's and their family members but the views of people with Parkinson's also differed, particularly whether information should focus on current symptoms alone or include the future. Some took the approach of "*the less I know sometimes, the less the better, I think*" (11P. Female, Diagnosed 6 years), meaning support groups were avoided out of fear of being exposed to people with advanced symptoms. For others, awareness of future possibilities helped:

*"You need to, somebody to tell you and to explain things to you before they get bad and before they're not–which is the things would be more helpful."* (3P. Female, Diagnosed 8 years)

Participants more commonly described wanting practical and positive information to help manage day-to-day matters and to respond to arising symptoms or situations. Discussions about future long-term care or end of life decisions were not reflected in the interviews, and coping appeared to be enabled by focusing on what it was possible to control:

"*Other people, they don't realise sometimes how you don't want to hear, like, people will say, 'oh, what will you do when it gets to this?' [. . .] I know you've got to make precautions for the future, but . . ..*" (06C. Daughter of male, Diagnosed < 1 year)

**Finding a place in the healthcare system.** Symptom progression meant increased reliance on healthcare providers, at which point some participants felt that they did not fit into the healthcare system, as Parkinson's was not always understood by the healthcare providers they interacted with. Many considered greater continuity of care and understanding of Parkinson's were required in primary and secondary healthcare and social care:

"*You can go to the doctor's on four different occasions and see four different doctors. So, we would like to think we have some sort of priority, it's nice to know you're going to be met by somebody you have some confidence in.*" (13P Male, Diagnosed 6 years)

Admission to hospital following acute episodes, or a care home for respite, occurred when symptoms became more complex, often accompanied by a marked deterioration and difficulty

coping at home. When moving from home to hospital people described feeling that their Parkinson's specific needs were not fully met:

> "We told this staff nurse that your regime of four times a day with tablets is no good for my mum, it's six times a day, or in actual fact, seven. That didn't happen at all." (14C. Son of female, Diagnosed 14 years]

Similarly, there were complaints that despite the creation of individual care plans by nursing and care staff there was limited understanding of the fluctuations of Parkinson's symptoms by non-specialist staff working in non-specialist environments:

> "People don't understand the Parkinson's at all, especially normal wards [general medical wards]. [. . .] You know they fill out these questionnaires and they say what you can do for yourself. And, you know, I've ticked like, 'going to the bathroom, I may need help, getting dressed, I need help.' [. . .] the nurses will see that you're young and you look fine, so you don't need any help. There will be times where they'll leave the lunch there, but sometimes you can't cut the food up yourself." [03P. Female, Diagnosed 8 years]

With progression of symptoms participants spoke of the increasing benefit from specialist advice, including neurologists, Parkinson's nurses, physiotherapists and speech and language therapists. The perceived variation in provision and access to specialist services was however criticised:

> "But the main problem is the diversity of what's out there and putting it all together. I mean, I've spoken to people who've never had any contact with physiotherapy, who've never had any contact from occupational therapists. I mean, I didn't have contact from occupational therapists until I'd been assessed for medical retirement, which was nine years in." (05P. Male, Diagnosed 18 years)

**Increasing dependency.** The final transition identified was as symptoms increased and participants experienced a significant decline in functional ability. It was identified as: *changes in roles and relationships* and becoming *increasingly dependent*.

**Changes in roles and relationships.** As symptoms increased participants spoke about reduced independence, subsequent role changes and increasing reliance on others for practical support, facilitation of social activities and assistance with health and care, with co-resident spouses and family members invariably taking on more responsibility. Transition within roles and renegotiation started with supervising, monitoring and supporting activities:

> "And she watches me while I'm getting out of the bath and getting into the bath, to make sure that I don't fall down." (06P. Male, Diagnosed < 1 year)

Renegotiation also took place through a process of sharing activities, where the person with Parkinson's continued to contribute in ways that were still possible alongside existing support:

> "If she was on her own, she'd struggle to cook the sort of meal that she would like [. . .] So now she sits out there on a stool and says, 'Right you set that timer for that time and this timer for that time and this and that.'" (07C. Husband of 74 year old female, diagnosed 8 years)

However, over time, physical and cognitive decline meant tasks were relinquished, such as managing medications or finances:

*"I just do the housework or sort out the house, talk to the insurance assessor, talk to these people, those people, or make his appointments and remind him of the appointments. [. . .] The total burden of running the house is on me." (*18C. Wife of male, Diagnosed 5 years)

As attention increasingly focussed on the needs of the person with Parkinson's, spouses often surrendered their own needs to accommodate reduced ability or to capitalise on good periods. They also became conscious of the transition from 'spouse' to 'carer', exacerbated by symptoms such as nocturia and sleep disturbance often leading to sleeping in separate rooms, resulting in loss of intimacy:

*"I just do feel like 'the carer' [. . .]. Well, we're not really physically intimate now, I think. He probably just sees me more as his carer."* (05C. Wife of male, Diagnosed 18 years)

Although co-resident family members such as spouses provided substantial care and support, there were also instances where role transitions occurred to provide further support from non-resident family members:

*"My granddaughter comes in once a fortnight and cleans the house from top to bottom for me."* (04P. 71 year old male)

**Increasingly dependent.** Declining abilities gradually limited independence, with activities and routine tasks taking longer and requiring careful use of energy:

*"For a normal person, they have their energy and use it everywhere. With Parkinson's, I use my energy wisely. I think to myself, what's more important for me to do today. And I'll put my energies there."* (03P. Female, Diagnosed 8 years)

Several participants spoke knowledgably and insightfully about the debilitating symptoms leading to their dependence. They described increasing unpredictability of symptoms and 'on'/ 'off' periods, when medications became less effective and resulted in tiredness, slowness and impossibility of doing things and therefore increased reliance on others:

*"I run out of dopamine. And once you run out of dopamine, the whole thing goes haywire. It's very difficult to do anything then. [. . .] So from 3 o'clock onwards, we're on a downward slope to bedtime which is at 10 or 11 o'clock at night."* (12P male, Diagnosed 3 years)

Although many changes took place over time, a diagnosis of dementia or discharge from hospital could highlight the extent of increasing difficulty and decline. At such times increased recognition of the need for additional support often triggered the involvement of services such as home care, and members of the multi-disciplinary team:

*"It's progressively got more challenging with her deteriorating condition, which is now Parkinson's dementia as well [. . .] that's when I think it really hit home because she was, I think, a bit in denial up until then. And then shortly after that, we were seen, or she was*

*seen by a mental health nurse, a dementia nurse."* (10C. Husband to female, Diagnosed 2 years)

Although increasingly less able and dependent on family members and external support agencies participants described thinking optimistically, not worrying about the unknown, gratitude for the past, acknowledging positive aspects of life, taking one day at a time, and making the most of the good days as ways of coping with worsening symptoms:

*"I've been very lucky. I had a lovely wife, I've got lovely children. I've got lovely friends. What more can you want? I've got a lovely flat."* (8P. Male Diagnosed 18 months)

## Discussion

We identified three key points of transition, 'being told you are a person with Parkinson's' (early), 'Living with Parkinson's' (mid) and 'Increasing dependency' (decline). The data suggest that transitions were also mirrored among family members who both witnessed transitions in their relative with Parkinson's and experienced their own changes of identity. Previous work has considered the transitions in the later stages of Parkinson's [24] and those experienced by couples in the context of support [25]. This current paper broadens the view of transitions with each theme as a point of transition across the evolving course of Parkinson's and to all aspects of living with the condition.

Parkinson's affects all areas of daily life, requiring adaptation in internal and external life domains which have to be reframed over time [26] reflecting the ongoing transitions described in our findings and earlier work [27]. The concept provides a framework for the multi-dimensional, multi-layered processes of transition currently describing other chronic conditions [28]; providing a holistic understanding of Parkinson's. Whilst some of the challenges of having Parkinson's have been reported elsewhere [12, 29–31] to our knowledge this is the first study to distinguish these various transitions in Parkinson's in this way.

### Being told you are a person with Parkinson's (early)

The first point of transition was identified as when participants were told they have a chronic neurodegenerative disease, and our findings support the literature in that the diagnosis of Parkinson's could be shocking and take time to be confirmed [31–33]. This paper further adds that the quality of diagnosis delivery was seen as important, including compassion and reassurance [31], the opportunity to ask questions [32], but importantly to also feel that investigations had been thorough. Whilst participants in this study did not generally report dissatisfaction about the way diagnosis was delivered, as found in other studies [34], participants from across a range of disease durations vividly recalled receiving their diagnosis, suggesting that the impact was still felt and resounded across the disease course, potentially informing adjustment. This is of relevance as studies suggest that how the diagnosis is delivered can have a negative long-term impact on quality of life [35] and satisfaction with care [21]. Improved pre and post-diagnostic care including clear, empathetic explanations of the diagnosis process could be of value (Box 1) and there is learning from other long-term conditions, such as dementia, where there are aspirations to 'diagnose well' to mitigate the impact of diagnosis delivery [36].

## Box 1. Clinical implications

| Implications informed by the over-arching concept of transitions: |
| --- |
| ➢ The concept of transitions could act as a map to identify times of significant adaption and change, of what is important to the individual, and which support may be needed |
| ➢ Transitions could be used to encourage and engage in discussions acting as a prompt/guide |

Implications informed by themes and sub-themes

*Theme*: *Being told you are a person with Parkinson's (early phase)*:

| Sub-theme | Potential strategies to address challenges identified in sub-theme |
| --- | --- |
| Receiving and accepting a diagnosis; Navigating reactions | ➢ Pre- and post-diagnostic care could include clear, empathetic explanations of the diagnostic process |
| | ➢ Consider facilitation of meeting with others at the same age and stage, i.e., signposting to a young-onset PD support group or diagnosis specific group |

*Theme*: *Living with Parkinson's (mid* phase)

| Sub-theme | Potential strategies to address challenges identified in sub-theme |
| --- | --- |
| Changing social interactions and maintaining sense of self | ➢ Ongoing adaptive facilitation by health professionals to help those with Parkinson's and family members maintain and promote social engagement for example coaching to improve self-efficacy and manage changes in social roles |
| | ➢ Individuals and health professionals should focus on assets, such as abilities, strengths, existing support networks and meaningful activities rather than responding to functional deficits alone |
| | ➢ Facilitate enabling individuals to recognise their needs, make informed choices, and prepare for the incorporation of future changes that will be increasingly demanded by Parkinson's |
| Information: wanting to know but not wanting to know | ➢ Bespoke, tailored information for those with Parkinson's and family members |
| | ➢ Facilitating collaborative approaches between family members in preparation for when discussion about long-term future and end of life care is needed |
| Finding a place in the healthcare system | ➢ Promote awareness of such campaigns such as Parkinson's UK 'Get it on time' [64] that works to ensure needs are met within the healthcare system |
| | ➢ Support with navigating the healthcare system |
| | ➢ Training of non-specialist Healthcare Professionals about Parkinson's |
| | ➢ Improved continuity in care |

*Theme*: *Increasing dependency (decline phase)*.

| Sub-theme | Potential strategies to address challenges identified in sub-theme |
| --- | --- |
| Changes in roles and relationships | ➢ Individual and ongoing support for spouses and family carers |
| Becoming increasingly dependent | ➢ Important to increasingly involve those with Parkinson's and families in all consultations and decisions about delivery of care, to acknowledge their experiences and facilitate collaborative communication and working between family members |
| | ➢ Agility of healthcare professionals to respond to changes in order to offer appropriate, timely support for areas of life that are important to the individual |

Participants with Parkinson's described their psychosocial adjustment to the diagnosis as both an internal and external process, including navigating others' reactions. Concerns around

telling family, friends and employers about their diagnosis was managed by either being open, seen as a positive way of helping people understand Parkinson's [37]; or keeping their diagnosis hidden due to either personal reluctance to accept the diagnosis, sometimes because of stigma [14, 38], or legitimate concerns about employment [39]. As supported by the literature [33, 40, 41] our findings suggest that experiences differ by age and stage of life. Peer-led approaches have been shown to improve activity and well-being [42, 43], however there can be ambivalence in attending such groups, possibly out of fear of encountering others with more advanced conditions [24]. The facilitation of meeting with others at the same age and stage, for example signposting to a young-onset, or recently diagnosed Parkinson's support group, for example, the First Steps programme [42], could be of value in managing this first transition as identified in this study (Box 1).

## Living with Parkinson's (mid phase)

The 'mid' phase of 'Living with Parkinson's' was when symptoms increasingly and to varying degrees affected mobility, memory, mood and speech, presenting challenges to remaining socially connected [44, 45]. The importance of social connectedness with multiple positive influences, including providing social identity and a sense of belonging, life satisfaction and well-being [12, 46] suggests the need for ongoing adaptive facilitation for social engagement given the progressive nature of Parkinson's. Those specifically in the mid-phase may therefore benefit from support in navigating changing social roles, for example coaching to improve self-efficacy and manage changes in social roles (Box 1).

Despite the constraint of declining capacity participants demonstrated the desire to preserve abilities, connections and normality. This was shown for example, by creatively adapting hobbies and contact with existing social groups, and sharing tasks with spouses rather than relinquishing tasks. This desire to preserve the pre-Parkinson's self is reflected elsewhere [30, 47, 48]. Studies have also shown the importance of maintaining personal resources, function and mental health [49] and have discussed the value of support to preserve health and independence rather than focus on disability and dependence [50]. This study adds to the knowledge by showing that such an asset model to promote individual self-esteem and coping abilities [51] and preserve strengths, existing support networks and meaningful activities [12] could be further applied by health and care practitioners. This could be considered through the application of holistic interventions and social prescribing to promote well-being alongside 'reactive' responses to functional 'deficits' at the mid-phase transitions identified here (Box 1). In addition to preserving existing assets health professionals could also support individuals to recognise their needs, make informed choices, and prepare for the incorporation of future changes that will be increasingly demanded by Parkinson's (Box 1) as suggested for managing transitions in other chronic conditions [52]. Similarly, supporting those with Parkinson's to preserve a positive mindset and determination may be helpful as this has been shown to contribute to positive adjustment in Parkinson's [37].

Information was important in navigating times of transition for both those with Parkinson's and family members, particularly practical and positive information to help control and cope with current daily life [11, 12, 41]. Views however varied on the scope and depth of content and timing of delivery, supporting, the need for information to be bespoke [11, 26], not least given the heterogeneity of Parkinson's. Findings extend this knowledge by suggesting that the management of information needs careful discussion especially for family members who may feel the need to search for information to prepare for the future, whilst wishing to protect the person with Parkinson's. Therefore, finding ways of facilitating more collaborative

approaches between family members during the transition towards increasing dependency may be indicated, particularly in preparation for discussions about long-term future and end of life care [53] (Box 1).

Symptom progression meant increased reliance on healthcare providers, however participants did not always 'find their place' in the healthcare system as Parkinson's was not always understood by the non-specialist healthcare providers they interacted with, most notably for symptoms requiring hospital admission [54, 55]. Participants spoke of the increasing benefit from specialist advice as their symptoms deteriorated or became more complex [26, 27, 29] which has implications for workforce planning.

## Increasing dependency (decline)

In this phase, significant increases in debilitating symptoms led to substantial reliance on others, especially co-resident spouses and family members, as reflected in a Carers UK report [56]. Transition to dependency included renegotiation of roles, with spouses often surrendering their own needs to accommodate their partners' reduced ability or to capitalise on good periods, eventually changing from 'spouse' to 'carer' [25]; with subsequent strain as reflected elsewhere [6, 7, 29, 57]. These findings support the need for ongoing support for family carers who provide physical, emotional and social care in the home to those with advancing Parkinson's (Box 1). This is highly relevant for times marking significant deterioration, for example a diagnosis of dementia or admission and discharge from hospital resulting in increased commitments from spouses or other family carers.

Advancing symptoms, declining abilities and significant losses increasingly limited independence and affected the entirety of everyday life [27], impacting quality of life and life satisfaction of those with Parkinson's [5, 58]. Despite declining ability participants spoke with insight and knowledge about their management of troublesome symptoms, for example fatigue, and also about medications and their effectiveness, especially in relation to unpredictable symptoms and 'on'/'off' periods, also described in previous work [59]. In the later stages of Parkinson's such a sense of autonomy and self-efficacy have been shown to be associated with life satisfaction [60].

Despite increasing dependency and isolation, both those with Parkinson's and family members described coping strategies such as focusing on the present by taking one day at a time and making the most of the good days, feeling gratitude for the past, and acknowledging positive aspects of life.

## Strengths and weaknesses

Our sample included perspectives from people with Parkinson's and family members, who were socially diverse, recruited from inner city, suburban and rural settings. A breadth of Parkinson's duration was represented and having more men than women is consistent with the overrepresentation of men diagnosed with Parkinson's [1]. The study team spanned a range of clinical and academic expertise including neurology, primary care, gerontology, nursing, social care, psychology and a person with lived experience of Parkinson's, which aided interpretation of findings and the consideration of multiple perspectives and ideas. The analysis would have benefited from inclusion of a current family carer, although one member of the research team had previously been a carer of someone with Parkinson's.

However, ethnic diversity in the sample is limited, despite efforts to recruit from services with higher prevalence of ethnic minority groups. Single people living alone were also underrepresented and may have different experiences to those living with others. When considering the transferability of findings, despite international studies describing similar Parkinson's

symptoms and impacts [31, 61, 62], it is important to acknowledge that health services have regional and international variations [63] potentially changing the experiences of those with Parkinson's.

## Clinical implications

As described throughout the discussion, a range of potential clinical implications have been identified at each transition point and these are presented in Box 1.

## Future research

Longitudinal exploration of the experiences of those with Parkinson's and carers would be of value to understand transitions over a prolonged period of time, for example using case study methods. Further detailed exploration of each phase of transition for both those with Parkinson's and the carer would be of value, for example examining stages of grief associated with managing each phase. It is important to explore which interventions and strategies over time are successful at addressing the challenges within the transitions identified here. For example, whether the reluctance to read about later stages of the condition affects care planning, particularly for those who may develop cognitive impairment, or is helpful in reducing anxiety and improving quality of life.

## Conclusion

Those with Parkinson's, and those closest to them, adapt their internal and external life domains over the trajectory of the condition from diagnosis through to decline and increasing dependence. This understanding is important to healthcare professionals, particularly non-specialists, so as to be aware of the individual's changing needs and to support the necessary adaptations at key transitions such as diagnosis, changes in social connections, and increased use of expertise so that timely, comprehensive, holistic and individualised support can be put in place to maintain well-being.

## Supporting information

**S1 Checklist.**
(PDF)

**S1 File.**
(DOCX)

## Acknowledgments

We thank all participants who took part in this study, offered their time and shared their experiences with the team. We would like to thank Tanisha DeSouza who helped to collect data and assisted with preliminary analysis.

## Author Contributions

**Conceptualization:** Kate Walters, Anette Schrag.

**Data curation:** Joy Read, Jennifer Pigott, Nathan Davies.

**Formal analysis:** Joy Read, Rachael Frost, Kate Walters, Remco Tuijt, Jill Manthorpe, Bev Maydon, Jennifer Pigott, Anette Schrag, Nathan Davies.

**Funding acquisition:** Kate Walters, Jill Manthorpe, Bev Maydon, Anette Schrag, Nathan Davies.

**Methodology:** Kate Walters.

**Project administration:** Joy Read, Anette Schrag.

**Supervision:** Rachael Frost, Kate Walters, Anette Schrag, Nathan Davies.

**Writing – original draft:** Joy Read, Rachael Frost, Nathan Davies.

**Writing – review & editing:** Joy Read, Rachael Frost, Kate Walters, Remco Tuijt, Jill Manthorpe, Bev Maydon, Jennifer Pigott, Anette Schrag, Nathan Davies.

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
