## [Decision Letter · Decision Letter 0]

12 Jan 2022

PONE-D-21-37562Transitions and challenges for people with Parkinson’s and their family members: A qualitative study

PLOS ONE

Dear Dr. Davies,

Thank you for submitting your manuscript to PLOS ONE. After careful consideration, we feel that it has merit but does not fully meet PLOS ONE’s publication criteria as it currently stands. Therefore, we invite you to submit a revised version of the manuscript that addresses the points raised during the review process.

Both reviewers have a number of positive comments regarding the potential of your manuscript of making a contribution to the literature but also highlight a number of issues with your method of analysis in terms of wanting some more details on the steps undertaken. I also agree with Reviewer #1 to include the interview guide and to also submit the COREQ with a revised manuscript.

We look forward to receiving your revised manuscript.

Kind regards,

Sander L. Hitzig

Academic Editor

PLOS ONE

Journal Requirements:

2.  Please include additional information regarding the interview guide used in the study and ensure that you have provided sufficient details that others could replicate the analyses. Please also include a copy as Supporting Information.

(NO authors have competing interests)

Reviewers' comments:

Reviewer's Responses to Questions

**Comments to the Author**

1. Is the manuscript technically sound, and do the data support the conclusions?

Reviewer #1: Yes

Reviewer #2: Partly

2. Has the statistical analysis been performed appropriately and rigorously? 

Reviewer #1: N/A

Reviewer #2: N/A

3. Have the authors made all data underlying the findings in their manuscript fully available?

Reviewer #1: No

Reviewer #2: No

4. Is the manuscript presented in an intelligible fashion and written in standard English?

Reviewer #1: Yes

Reviewer #2: Yes

5. Review Comments to the Author

Reviewer #1: The authors present a qualitative project involving people with Parkinson’s and caregivers and identifying themes relating to transitions over the course of PD. The themes are very resonant with my clinical experiences in a specialty Parkinson clinic and the discussions that I have with my patients and families about changing experiences over time. This will be an important contribution to the literature, especially for clinicians who may have less experience with this on a daily basis.

The explanation of what the authors mean by “transition” in this context is important and it is good that they included it and distinguished it from the more common use of the word to mean changes in location of care.

Justifications should be provided for the selection of this qualitative analysis approach.

The semi-structured interview should be included as supplemental materials.

Authors should provide one of the checklists for qualitative research reporting, such as COREQ.

In the analysis section of the methods, it is unclear how many individuals performed the initial coding frame. Was this a single individual, who then discussed the proposed frame with the team? Or was there dual coding? Also, “read a proportion of the transcripts” is very passive. Was the team actively involved in refining the coding frame and identifying themes and subthemes? Were themes and subthemes revised iteratively over time? When the final frame/codebook was determined, were the original transcripts re-reviewed using the new codebook? It says the frame was applied to remaining interviews but it is important to re-review the initially assessed transcripts once the final theme is determined, for differences that might occur because of changes in the codebook.

Discussion of results with a representative with lived experience of Parkinson is a strength. Ideally this would be with a person with PD and a caregiver, since both groups were represented in the study.

I could not find a legend for Figure 1 and while I like the figure overall, it would benefit from some explanatory text. For example, I understand the trajectory because it is something that I often draw in clinic, but right now the declining line superimposed on the figure is somewhat confusing and distracting and it is possible that the non-specialist reader will not understand it without explanation. Also, the current formatting of the figure is somewhat confusing (at least how it appears in the PDF for review). The line with the acute on chronic declines is on the top, interrupting a bunch of the text, while the arrows explaining the sudden dips are at the bottom. Even as a specialist, this took me a little while to figure out (e.g., that the top line and the bottom arrows were related). I suggest putting the trajectory on its own – still in the figure, but not superimposed on the text. In addition to adjusting the layout of the figure (or how it is seen in the review PDF), a text legend will help explain what the figure is showing, particularly to those who do not routinely draw similar graphics for patients and families in the clinic. The overall figure is nice with its combination of trajectory and themes, but it could be improved with some fairly minor edits.

It would be helpful to define some of the terms that are not universal. For example, “general ward staff” and “normal wards” (page 15) are not vocabulary typically used in the U.S. Can the authors briefly clarify (even parenthetically) this context for readers from different healthcare systems?

The discussion does not always seem to draw specifically from the results. For example, on page 19, the authors say, “The impact of the way in which a diagnosis is delivered resounded across the disease course,” but this was not clearly described in the results, nor was it a specific theme or subtheme outlined in the text or figure. The subsequent statements about diagnosing well also need references if maintained. The disconnect between the results and the discussion persists through most of the discussion. Typically the discussion section puts the new results into context, but in this manuscript there is very little discussion of how current findings are consistent with or different from what is currently published in the literature. The discussion would benefit from some rewriting, where current results are placed into context of the published literature and then the authors make suggestions regarding how clinics/clinicians can address the challenges identified in the study. The discussion also seems to ramble a bit at times. It might be helpful to create some subheadings in the discussion to help the reader understand the points the authors are trying to emphasize.

Multiple quotes describe the effect of motor fluctuations in the “mid” period, but this subtheme seems to be missing from the analysis. Given that multiple quotes describe the impact of this (and presumably there are others not included in the manuscript itself), why is this not part of the subthemes relating to the middle phase? There is at least one recent paper describing many themes that overlap with what the participants in this study describe (doi:10.1212/CPJ.0000000000000921). There are also potentially relevant references not currently included, such as studies about early stages/diagnosis (e.g. https://doi.org/10.1177/1742395317694699).

Limitations should include uncertain generalizability to other healthcare contexts, though the authors can also point out where current findings are similar to results reported in research performed in other international locations.

Box 1 includes exact language for some subthemes but no for others. It would be clearer if the box specifically labeled each subtheme with consistent language and then presented potential strategies to help that phase/subtheme. The columns in the box should also be labeled. Presumably the first column references the subtheme and the second column represents potential strategies to help the challenges represented in the subtheme? There might also be a third column with references to support the suggested strategies, since this goes beyond what was studied (the interviews seem to focus on experiences, not helpful strategies that participants employed or received). Box 1 should also be integrated better with the discussion.

Suggest removing the sentence that findings might be of relevance to other degenerative conditions. It is likely that these other conditions have key differences in phases that would not be reflected in the current study. For example, people with MS may be affected at a younger average age, people with dementia are dependent at diagnosis, and people with HD have the challenge of living within a family where others are also likely to be affected and where they are likely to have seen the impact of the disease for years (in contrast to current findings, where people avoided support groups so they did not have to see later stages). Thus, this statement is likely incorrect but also largely irrelevant to the importance of the findings to care for people with PD and their families.

The future research section is underutilized. What about research looking at stages of grief involved with coping at each phase? What about caregiver experiences of the phases (despite interviewing caregivers, much of the framework of phases seems more patient-focused)? What exactly do the authors mean by “interventions”? Interventions to address challenges? The authors also seem to pre-suppose that reluctance to read about PD is a bad thing. What if it improves quality of life and lessens anxiety?

In the conclusion the authors say that “the concept of transition was a meaningful way of conceptualizing the changes psychologically…” but it is unclear – meaningful to who? Meaningful to the authors? It sounds like this was a concept developed out of the transcripts. If that is correct, then the participants could not have endorsed this as meaningful. If this framework was pre-supposed, that should be made clear in the methods and the analysis. Having the semi-structured questionnaire would also help understand this.

There is no data availability statement within the manuscript. In the questions, the authors state, "All relevant data are within the manuscript and its Supporting Information files." I cannot find any supporting information files available to review. While typically the transcripts themselves are not included as supplemental files as the whole of a transcript might be identfiable even if specific identifiers are redacted, qualitative work published in PLoS One has included supplemental materials like the code book with the associated quotes for each theme and subtheme (rather than just the select quotes that are in the manuscript). Currently there is no data availability that I can tell - nothing included in supplemental materials, in a data respository, or on request.

Reviewer #2: This was a well-written and interesting paper which considers the challenges of and transitions between the different stages of Parkinson’s. I have a few comments, mainly regarding the description of the method and the addition of further current literature in the discussion. I have selected "partly" to question 1 because I think the description of the data analysis needs to be improved. I do not have any concerns about the results or conclusions - I just think the description of the process of analysis needs to be clearer so one can be confident it is "technically sound".

Abstract

As the authors indicate in the introduction, the term transition is often used to indicate a move between different services/types of treatment etc, as well as being used in the sense the authors mean here, more a transition between disease states/ways of living. I am wondering therefore if this could be made clearer in the title and/or abstract. For example the objective is currently: “To understand the transitions experienced by people with Parkinson’s and their family members living in the community.” However, this does not make it clear what is meant by transition – so could this be explained a little more? It would also be helpful in the abstract to know which subthemes belong to which themes.

Introduction

This is mainly clear – with just some queries about the final paragraph, covered below.

Method

A little more explanation and coherence is needed regarding the method. The authors say that they conducted thematic analysis using Braun & Clarke’s (2006) approach. However, Braun & Clarke have since developed and refined their method and tend now to refer to “reflexive thematic analysis” (e.g. Braun, V. & Clarke, V. (2019) Reflecting on reflexive thematic analysis, Qualitative Research in Sport, Exercise and Health, 11:4, 589-597, DOI: 10.1080/2159676X.2019.1628806). As they outline, there are actually various types of thematic analysis so it is helpful to be more specific about which one is meant. There is a helpful summary of their development of the approach here: https://www.psych.auckland.ac.nz/en/about/thematic-analysis.html and in their new book (Braun, V. & Clarke, V. (2021). Thematic Analysis: A Practical Guide. Sage.) So I think the authors need to be more explicit whether they are using the inductive approach that Braun & Clarke outline (reflexive thematic analysis) or whether they are using a more deductive approach (e.g. codebook thematic analysis or framework analysis for example).

The introduction says that the “lens of transitions” (page 4) will be used to look at the “lived experience” and then later the same paragraph talks about the “framework of transitions”. It also says that “mapping Parkinson’s onto transitions” (page 4) is a new approach. How was the mapping and using of a framework/lens actually implemented in practice? At what point was the lens introduced when conducting the data analysis? What did the mapping involve? Was the “framework of transitions” taken from previous literature or constructed for the current project? There is very little about this in the method. The method states that an “inductive approach” (page 5) was used and then later that “the concept of transitions was identified and applied” – so does this mean that the concept of transitions came from the data (i.e. when doing the inductive analysis, an overarching theme of transitions was seen) or that the project was about transitions specifically and that the interview schedule and analysis was conducted specifically focusing on transitions? This needs to be a bit clearer both in the introduction and method. The process of analysis then needs more explanation with appropriate methodological references.

As a more minor issue- in a couple of places the language used does not quite align with the Braun & Clarke conceptualisation of the method. For example, Braun & Clarke argue strongly against using the word “emerge” for themes (as is used here on page 4: “no new themes emerged”) and similarly are critical about the concept of saturation (Braun, V. & Clarke, V. (2021) To saturate or not to saturate? Questioning data saturation as a useful concept for thematic analysis and sample-size rationales, Qualitative Research in Sport, Exercise and Health, 13:2, 201-216, DOI: 10.1080/2159676X.2019.1704846). Thus a little more is needed here (with appropriate references) to indicate how “saturation” was conceptualised and utilised.

Page 11: The topic guide was developed with reference to “the study objectives and literature” – could the citations of the relevant literature be included?

Pages 11-12: “The separate code lists were compared and reviewed to create an initial coding frame. The coding frame was discussed with other members of the team (anon), who read a proportion of the transcripts. The agreed coding frame was then applied to the remaining interviews”. How many transcripts were looked at first to form the initial coding frame?

Results

The three themes and subthemes seem coherent and informative and the thematic map (Figure 1) gives a helpful overview of the findings.

A perhaps minor point, but the quotations largely seem to focus on each phase (i.e. be static), rather than talk about the transition from one phase to the next, which is the focus, I think, of the current work? Is there anything that can be done in each theme to bring the nature of the transition (from one phase to the next) more to the fore, if this indeed is part of the intended focus? Or is each theme/phase itself seen as a transition? Perhaps this could this be clearer.

Discussion

There is a considerable body of (arguably) similar work in Parkinson’s that I think needs some consideration in the discussion. The Soundy et al. (2014) review is already included but there are also two more recent reviews, which I think could be relevant, both as a whole and the papers cited within: Rutten S, van den Heuvel OA, de Kruif A, et al. The subjective experience of living with Parkinson’s disease: a meta-ethnography of qualitative literature. J Parkinsons Dis.2021;11(1):139–151 and Wieringa G, Dale M, Eccles FJR. Adjusting to living with Parkinson's disease; a meta-ethnography of qualitative research. Disabil Rehabil. 2021:1-20. doi: 10.1080/09638288.2021.1981467. Epub ahead of print.

The Rutten et al. review is particularly relevant to the current paper as it discusses the changes over time. The following paper also perhaps has findings relevant to the current work: Vann-Ward T, Morse JM, Charmaz K. Preserving Self: Theorizing the Social and Psychological Processes of Living With Parkinson Disease. Qualitative Health Research. 2017;27(7):964-982. doi:10.1177/1049732317707494)

Page 19: “Parkinson’s affects all areas of life and acceptance has to be reframed over time (20) reflecting the ongoing transitions described in our findings.” Can this sentence be explained a little more?” Acceptance is only mentioned explicitly once in the results (as far as I can see) so it is not clear at the moment how the acceptance changing over time plays out in the results of the current paper. I think either this needs making more explicit throughout the results, or the way in which acceptance changes throughout the transitions needs to be explained more here in the discussion.

Page 19: “Whilst some of the challenges of having Parkinson’s have been reported elsewhere (12, 22-24) to our knowledge this is the first study to map them onto these various transitions across the Parkinson’s.” Again, linking to my comment above on the method – was the aim therefore to “map” difficulties onto the different phases of Parkinson’s?

One paragraph on page 19 focuses on the diagnosis phase. Several previous papers have also similarly considered the effects of receiving a diagnosis and could be cited including: Phillips, L.J. (2006). Dropping the bomb: The experience of being diagnosed with Parkinson’s disease. Geriatric Nursing, 27, 362-369. doi: 10.1016/j.gerinurse.2006.10.012 Pinder. (1992). Coherence and incoherence: doctors’ and patients’ perspectives on the diagnosis of Parkinson’s disease. Sociology of Health and Illness, 13, 1-23. doi: 10.1111/j.14679566.1992.tb00111.x and Warren, E., Eccles, F., Travers, V., & Simpson, J. (2016). The experiences of being diagnosed with Parkinson’s disease. British Journal of Neuroscience Nursing, 12, 288-296. doi: 10.12968/bjnn.2016.12.6.288.

The following review includes Parkinson’s and so also may be relevant: Anestis E, Eccles F, Fletcher I, French M, Simpson J. Giving and receiving a diagnosis of a progressive neurological condition: A scoping review of doctors' and patients' perspectives. Patient Educ Couns. 2020; 103(9):1709-1723. doi: 10.1016/j.pec.2020.03.023. Epub ahead of print.

Page 19: “The impact of the way in which a diagnosis is delivered resounded across the disease course and there is learning from other long-term conditions (LTC), such as dementia, where there are aspirations to ‘diagnose well’ to mitigate the impact of diagnosis delivery.” I think there should be a citation after this statement.

Finally, the following study may be a useful comparator? Bogosian, A., Morgan, M., Bishop, F. L., Day, F., & Moss-Morris, R. (2017). Adjustment modes in the trajectory of progressive multiple sclerosis: a qualitative study and conceptual model. Psychology & Health, 32(3), 343–360. https://doi-org.ezproxy.lancs.ac.uk/10.1080/08870446.2016.1268691

6. PLOS authors have the option to publish the peer review history of their article (what does this mean?). If published, this will include your full peer review and any attached files.

Reviewer #1: No

Reviewer #2: No

---

## [Author Response · Author response to Decision Letter 0]

11 Mar 2022

We would like to thank both reviewers and the editor for their positive review of our article and comments which we believe have strengthened our article. We have listed comments below and our responses. These have been highlighted in the revised manuscript.

Both reviewers have a number of positive comments regarding the potential of your manuscript of making a contribution to the literature but also highlight a number of issues with your method of analysis in terms of wanting some more details on the steps undertaken. I also agree with Reviewer #1 to include the interview guide and to also submit the COREQ with a revised manuscript.

1., Please ensure that your manuscript meets PLOS ONE's style requirements, including those for file naming. 

We trust the manuscript meets PLOS ONES style requirements

2., Please include additional information regarding the interview guide used in the study and ensure that you have provided sufficient details that others could replicate the analyses. Please also include a copy as Supporting Information.

We have made additional reference to the content of the interview guide within the manuscript i.e., by including examples of questions used – See page 5, lines 130-135. We have also submitted a copy of the interview guide as Supporting Information. 

We have added further information to the analysis section see page 6, lines 156-170.

3., Thank you for stating the following in your Competing Interests section: 

(NO authors have competing interests)

Thank you for this, we have declared in our cover letter we have no competing interests. 

The data is from interviews as part of a qualitative study. We do not have permission from participants to share their transcripts and data publicly, other than extracts for the purpose of publications. All relevant excerpts of data have been made available in the paper.

Reviewers' comments:

3. Have the authors made all data underlying the findings in their manuscript fully available?

Reviewer #1: No

Reviewer #2: No

See previous response to editors comment. 

5. Review Comments to the Author

Reviewer #1: The authors present a qualitative project involving people with Parkinson’s and caregivers and identifying themes relating to transitions over the course of PD. The themes are very resonant with my clinical experiences in a specialty Parkinson clinic and the discussions that I have with my patients and families about changing experiences over time. This will be an important contribution to the literature, especially for clinicians who may have less experience with this on a daily basis.

The explanation of what the authors mean by “transition” in this context is important and it is good that they included it and distinguished it from the more common use of the word to mean changes in location of care.

Justifications should be provided for the selection of this qualitative analysis approach.

We have added this to page 6-7, lines 162-170.

Q., Authors should provide one of the checklists for qualitative research reporting, such as COREQ.,

A completed copy of the COREQ checklist has been uploaded, and reference made within the text. 

Q., In the analysis section of the methods, it is unclear how many individuals performed the initial coding frame. Was this a single individual, who then discussed the proposed frame with the team? Or was there dual coding? Also, “read a proportion of the transcripts” is very passive. Was the team actively involved in refining the coding frame and identifying themes and subthemes? Were themes and subthemes revised iteratively over time? When the final frame/codebook was determined, were the original transcripts re-reviewed using the new codebook? It says the frame was applied to remaining interviews but it is important to re-review the initially assessed transcripts once the final theme is determined, for differences that might occur because of changes in the codebook.

Thank you for this point. The analysis section has now been substantially rewritten in order to address the points raised, specifically to clarify how many people created the initial coding frame, how it evolved further and was applied. The text also now includes a greater description of the multidisciplinary authors and their familiarity with the data and role in developing the themes and subthemes. See pages 6-7, lines 156-170.

Q., Discussion of results with a representative with lived experience of Parkinson is a strength. Ideally this would be with a person with PD and a caregiver, since both groups were represented in the study.

Thank you for identifying the strength of our work, we agree it could have been strengthened with the inclusion of a family carer in the discussions of the findings and we have added this to the strengths and weaknesses section, page 24, lines 617-619.

Q., I could not find a legend for Figure 1 and while I like the figure overall, it would benefit from some explanatory text. For example, I understand the trajectory because it is something that I often draw in clinic, but right now the declining line superimposed on the figure is somewhat confusing and distracting and it is possible that the non-specialist reader will not understand it without explanation. Also, the current formatting of the figure is somewhat confusing (at least how it appears in the PDF for review). The line with the acute on chronic declines is on the top, interrupting a bunch of the text, while the arrows explaining the sudden dips are at the bottom. Even as a specialist, this took me a little while to figure out (e.g., that the top line and the bottom arrows were related). I suggest putting the trajectory on its own – still in the figure, but not superimposed on the text. In addition to adjusting the layout of the figure (or how it is seen in the review PDF), a text legend will help explain what the figure is showing, particularly to those who do not routinely draw similar graphics for patients and families in the clinic. The overall figure is nice with its combination of trajectory and themes, but it could be improved with some fairly minor edits.

Thank you for pointing this out as it appears there was a formatting error on submission. The figure has been edited as suggested with addition of titles for both axis, re-arrangement of arrows, and a description of the figure added to the text. See page 9, lines 197-199.

Q., It would be helpful to define some of the terms that are not universal. For example, “general ward staff” and “normal wards” (page 15) are not vocabulary typically used in the U.S. Can the authors briefly clarify (even parenthetically) this context for readers from different healthcare systems?

We have now added an explanation of what the participant statement refers to non-specialist staff in a non-specialist environment. See page 16, lines 388-390. 

Q., The discussion does not always seem to draw specifically from the results. For example, on page 19, the authors say, “The impact of the way in which a diagnosis is delivered resounded across the disease course,” but this was not clearly described in the results, nor was it a specific theme or subtheme outlined in the text or figure. The subsequent statements about diagnosing well also need references if maintained. 

The discussion has now been amended so that all the results are more clearly integrated throughout. The resonance of the diagnosis is not considered to be a specific sub-theme, instead it is part of the existing sub-theme where a diagnosis is sought, delivery is commented upon, and the impact of this as described as resonating across the disease course. We have expanded and clarified the diagnosis section and added references in the discussion, which also addresses reviewer 2’s comments.

 Q., The disconnect between the results and the discussion persists through most of the discussion. Typically the discussion section puts the new results into context, but in this manuscript there is very little discussion of how current findings are consistent with or different from what is currently published in the literature. The discussion would benefit from some rewriting, where current results are placed into context of the published literature and then the authors make suggestions regarding how clinics/clinicians can address the challenges identified in the study. The discussion also seems to ramble a bit at times. It might be helpful to create some subheadings in the discussion to help the reader understand the points the authors are trying to emphasize.

Thank you for this important point, the discussion has been amended to better integrate the findings into the discussion and into the context of the existing literature, and link to box 1 where suggestions are made regarding how clinicians can address the identified challenges. 

Q., Multiple quotes describe the effect of motor fluctuations in the “mid” period, but this subtheme seems to be missing from the analysis. Given that multiple quotes describe the impact of this (and presumably there are others not included in the manuscript itself), why is this not part of the subthemes relating to the middle phase? There is at least one recent paper describing many themes that overlap with what the participants in this study describe (doi:10.1212/CPJ.0000000000000921). There are also potentially relevant references not currently included, such as studies about early stages/diagnosis (e.g. https://doi.org/10.1177/1742395317694699).

For our analysis motor fluctuation was not an independent sub-theme however along with other Parkinson’s symptoms it contributed to our understanding and development of sub-themes. 

Q., Limitations should include uncertain generalizability to other healthcare contexts, though the authors can also point out where current findings are similar to results reported in research performed in other international locations.

The strengths and weaknesses section, page 24-25, lines 623-627, now acknowledges that despite international studies reporting similar findings about Parkinson’s symptoms, health services have regional and international variations potentially changing the experiences of those with Parkinson’s. 

Q., Box 1 includes exact language for some subthemes but no for others. It would be clearer if the box specifically labeled each subtheme with consistent language and then presented potential strategies to help that phase/subtheme. The columns in the box should also be labeled. Presumably the first column references the subtheme and the second column represents potential strategies to help the challenges represented in the subtheme? There might also be a third column with references to support the suggested strategies, since this goes beyond what was studied (the interviews seem to focus on experiences, not helpful strategies that participants employed or received). 

Thank you for this observation. Box 1 now reflects the exact language of each sub-theme and the columns are now labeled. 

Q., Box 1 should also be integrated better with the discussion.

References to box one are now included throughout the discussion. 

Q., Suggest removing the sentence that findings might be of relevance to other degenerative conditions. It is likely that these other conditions have key differences in phases that would not be reflected in the current study. For example, people with MS may be affected at a younger average age, people with dementia are dependent at diagnosis, and people with HD have the challenge of living within a family where others are also likely to be affected and where they are likely to have seen the impact of the disease for years (in contrast to current findings, where people avoided support groups so they did not have to see later stages). Thus, this statement is likely incorrect but also largely irrelevant to the importance of the findings to care for people with PD and their families.

We agree and we have removed this statement. 

Q.,The future research section is underutilized. What about research looking at stages of grief involved with coping at each phase? What about caregiver experiences of the phases (despite interviewing caregivers, much of the framework of phases seems more patient-focused)? What exactly do the authors mean by “interventions”? Interventions to address challenges? The authors also seem to pre-suppose that reluctance to read about PD is a bad thing. What if it improves quality of life and lessens anxiety?

The future research section has now been expanded to include the suggested areas for additional exploration. An expanded description of ‘interventions’ as a means of addressing the challenges is now included. See page 25, lines 634-642.

Q., In the conclusion the authors say that “the concept of transition was a meaningful way of conceptualizing the changes psychologically…” but it is unclear – meaningful to who? Meaningful to the authors? It sounds like this was a concept developed out of the transcripts. If that is correct, then the participants could not have endorsed this as meaningful. If this framework was pre-supposed, that should be made clear in the methods and the analysis. Having the semi-structured questionnaire would also help understand this.

The concept of transitions was developed after reading through the transcripts not prior to data collection. We have removed this statement from the conclusion and elsewhere. 

Q., There is no data availability statement within the manuscript. In the questions, the authors state, "All relevant data are within the manuscript and its Supporting Information files." I cannot find any supporting information files available to review. While typically the transcripts themselves are not included as supplemental files as the whole of a transcript might be identfiable even if specific identifiers are redacted, qualitative work published in PLoS One has included supplemental materials like the code book with the associated quotes for each theme and subtheme (rather than just the select quotes that are in the manuscript). Currently there is no data availability that I can tell - nothing included in supplemental materials, in a data respository, or on request.

We have added a data sharing statement in response to the editor’s comment. 

Q., Reviewer #2: This was a well-written and interesting paper which considers the challenges of and transitions between the different stages of Parkinson’s. I have a few comments, mainly regarding the description of the method and the addition of further current literature in the discussion. I have selected "partly" to question 1 because I think the description of the data analysis needs to be improved. I do not have any concerns about the results or conclusions - I just think the description of the process of analysis needs to be clearer so one can be confident it is "technically sound".

Thank you for your positive comments about our paper, we are pleased you found it interesting. 

The description of the method and analysis has now been expanded upon, please see pages 6-7, lines 147-170.

Abstract

Q., As the authors indicate in the introduction, the term transition is often used to indicate a move between different services/types of treatment etc, as well as being used in the sense the authors mean here, more a transition between disease states/ways of living. I am wondering therefore if this could be made clearer in the title and/or abstract. For example the objective is currently: “To understand the transitions experienced by people with Parkinson’s and their family members living in the community.” However, this does not make it clear what is meant by transition – so could this be explained a little more?

The abstract has been amended accordingly to clarify the use of the word ‘transition’.

Q., It would also be helpful in the abstract to know which subthemes belong to which themes.

Numbers have been added so that it is clearer which subtheme is associated with the theme. Please see page 2, lines 41-46.

Introduction

Q.,This is mainly clear – with just some queries about the final paragraph, covered below.

This has been amended to clarify. Please see page 4, lines 91-92.

Method

Q., A little more explanation and coherence is needed regarding the method. The authors say that they conducted thematic analysis using Braun & Clarke’s (2006) approach. However, Braun & Clarke have since developed and refined their method and tend now to refer to “reflexive thematic analysis” (e.g. Braun, V. & Clarke, V. (2019) Reflecting on reflexive thematic analysis, Qualitative Research in Sport, Exercise and Health, 11:4, 589-597, DOI: 10.1080/2159676X.2019.1628806). As they outline, there are actually various types of thematic analysis so it is helpful to be more specific about which one is meant. There is a helpful summary of their development of the approach here: https://www.psych.auckland.ac.nz/en/about/thematic-analysis.html and in their new book (Braun, V. & Clarke, V. (2021). Thematic Analysis: A Practical Guide. Sage.) So I think the authors need to be more explicit whether they are using the inductive approach that Braun & Clarke outline (reflexive thematic analysis) or whether they are using a more deductive approach (e.g. codebook thematic analysis or framework analysis for example).

Thank you for this and we welcomed the new publications on this from Braun and Clarke. We believe we have followed the more codebook thematic analysis approach. Although a team approach was adopted this was not for a marker of quality as would be the case in coding reliability TA, but instead to widen our discussion, ideas and interpretation of the data. We have added detail of this type of TA to the analysis section and added more detail generally about our analysis. 

Q., The introduction says that the “lens of transitions” (page 4) will be used to look at the “lived experience” and then later the same paragraph talks about the “framework of transitions”. It also says that “mapping Parkinson’s onto transitions” (page 4) is a new approach. How was the mapping and using of a framework/lens actually implemented in practice? At what point was the lens introduced when conducting the data analysis? What did the mapping involve? Was the “framework of transitions” taken from previous literature or constructed for the current project? There is very little about this in the method. The method states that an “inductive approach” (page 5) was used and then later that “the concept of transitions was identified and applied” – so does this mean that the concept of transitions came from the data (i.e. when doing the inductive analysis, an overarching theme of transitions was seen) or that the project was about transitions specifically and that the interview schedule and analysis was conducted specifically focusing on transitions? This needs to be a bit clearer both in the introduction and method. The process of analysis then needs more explanation with appropriate methodological references.

The concept and relevance of transitions became apparent during the analysis. We have added this to the analysis section. 

Q., As a more minor issue- in a couple of places the language used does not quite align with the Braun & Clarke conceptualisation of the method. For example, Braun & Clarke argue strongly against using the word “emerge” for themes (as is used here on page 4: “no new themes emerged”) and similarly are critical about the concept of saturation (Braun, V. & Clarke, V. (2021) To saturate or not to saturate? Questioning data saturation as a useful concept for thematic analysis and sample-size rationales, Qualitative Research in Sport, Exercise and Health, 13:2, 201-216, DOI: 10.1080/2159676X.2019.1704846). Thus a little more is needed here (with appropriate references) to indicate how “saturation” was conceptualised and utilised.

We completely agree with the reviewer and have removed notion of emerge. We also agree the notion of saturation although has merits we feel is difficult, we were aware of the idea of information power when conducting our study and recruitment and were guided by the fact this was an explorative study, considered the quality and richness of our interviews, and have referred to this in our methods section (see page 4, lines 103-108). Towards the end of our data collection, we were aware as a team that the interviews were sufficient to answer our aims and objectives.

Q., Page 11: The topic guide was developed with reference to “the study objectives and literature” – could the citations of the relevant literature be included?

Some relevant references have now been added, however the citations are too wide to include all that would have informed our topic guide. 

Q., Pages 11-12: “The separate code lists were compared and reviewed to create an initial coding frame. The coding frame was discussed with other members of the team (anon), who read a proportion of the transcripts. The agreed coding frame was then applied to the remaining interviews”. How many transcripts were looked at first to form the initial coding frame?

The description in the analysis section (pages 6-7) has now been expanded upon to include such information. 

Results

Q., The three themes and subthemes seem coherent and informative and the thematic map (Figure 1) gives a helpful overview of the findings.

A perhaps minor point, but the quotations largely seem to focus on each phase (i.e. be static), rather than talk about the transition from one phase to the next, which is the focus, I think, of the current work? Is there anything that can be done in each theme to bring the nature of the transition (from one phase to the next) more to the fore, if this indeed is part of the intended focus? Or is each theme/phase itself seen as a transition? Perhaps this could this be clearer.

A clarifying sentence has been added, (see page 20 line 494) that each theme/phase is seen as a transition. 

Discussion

Q., There is a considerable body of (arguably) similar work in Parkinson’s that I think needs some consideration in the discussion. The Soundy et al. (2014) review is already included but there are also two more recent reviews, which I think could be relevant, both as a whole and the papers cited within: Rutten S, van den Heuvel OA, de Kruif A, et al. The subjective experience of living with Parkinson’s disease: a meta-ethnography of qualitative literature. J Parkinsons Dis.2021;11(1):139–151 and Wieringa G, Dale M, Eccles FJR. Adjusting to living with Parkinson's disease; a meta-ethnography of qualitative research. Disabil Rehabil. 2021:1-20. doi: 10.1080/09638288.2021.1981467. Epub ahead of print. 

The Rutten et al. review is particularly relevant to the current paper as it discusses the changes over time. The following paper also perhaps has findings relevant to the current work: Vann-Ward T, Morse JM, Charmaz K. Preserving Self: Theorizing the Social and Psychological Processes of Living With Parkinson Disease. Qualitative Health Research. 2017;27(7):964-982. doi:10.1177/1049732317707494)

Thank you for these very helpful references, which have been included in the reworked discussion. 

Q., Page 19: “Parkinson’s affects all areas of life and acceptance has to be reframed over time (20) reflecting the ongoing transitions described in our findings.” Can this sentence be explained a little more?” Acceptance is only mentioned explicitly once in the results (as far as I can see) so it is not clear at the moment how the acceptance changing over time plays out in the results of the current paper. I think either this needs making more explicit throughout the results, or the way in which acceptance changes throughout the transitions needs to be explained more here in the discussion.

The wording has been changed (page 20, lines 497-498) in order to add explanation to this point. 

Page 19: “Whilst some of the challenges of having Parkinson’s have been reported elsewhere (12, 22-24) to our knowledge this is the first study to map them onto these various transitions across the Parkinson’s.” Again, linking to my comment above on the method – was the aim therefore to “map” difficulties onto the different phases of Parkinson’s?

The concept and relevance of transitions became apparent during the analysis, which has been clarified in the methods section and discussion. 

Q., One paragraph on page 19 focuses on the diagnosis phase. Several previous papers have also similarly considered the effects of receiving a diagnosis and could be cited including: Phillips, L.J. (2006). Dropping the bomb: The experience of being diagnosed with Parkinson’s disease. Geriatric Nursing, 27, 362-369. doi: 10.1016/j.gerinurse.2006.10.012 Pinder. (1992). Coherence and incoherence: doctors’ and patients’ perspectives on the diagnosis of Parkinson’s disease. Sociology of Health and Illness, 13, 1-23. doi: 10.1111/j.14679566.1992.tb00111.x and Warren, E., Eccles, F., Travers, V., & Simpson, J. (2016). The experiences of being diagnosed with Parkinson’s disease. British Journal of Neuroscience Nursing, 12, 288-296. doi: 10.12968/bjnn.2016.12.6.288.

The following review includes Parkinson’s and so also may be relevant: Anestis E, Eccles F, Fletcher I, French M, Simpson J. Giving and receiving a diagnosis of a progressive neurological condition: A scoping review of doctors' and patients' perspectives. Patient Educ Couns. 2020; 103(9):1709-1723. doi: 10.1016/j.pec.2020.03.023. Epub ahead of print.

Thank you for these references, this is very helpful. In response to reviewer 1’s comments, we have expanded this section, including using these publications suggested above.

Q., Page 19: “The impact of the way in which a diagnosis is delivered resounded across the disease course and there is learning from other long-term conditions (LTC), such as dementia, where there are aspirations to ‘diagnose well’ to mitigate the impact of diagnosis delivery.” I think there should be a citation after this statement.

We have now added a reference. 

Q., Finally, the following study may be a useful comparator? Bogosian, A.,, Morgan, M., Bishop, F. L., Day, F., & Moss-Morris, R. (2017). Adjustment modes in the trajectory of progressive multiple sclerosis: a qualitative study and conceptual model. Psychology & Health, 32(3), 343–360. https://doi-org.ezproxy.lancs.ac.uk/10.1080/08870446.2016.1268691

Whilst a useful paper to be aware of we have not included reference to this paper because, as suggested by reviewer 1, people with MS may be affected at a younger average age and therefore report different experiences.

---

## [Decision Letter · Decision Letter 1]

8 Apr 2022

PONE-D-21-37562R1Transitions and challenges for people with Parkinson’s and their family members: A qualitative studyPLOS ONE

Dear Dr. Davies,

Thank you for submitting your manuscript to PLOS ONE. After careful consideration, we feel that it has merit, and the reviewers have pointed out that the manuscript is substantially improved but there remain a few minor issues that require some further attention.  I agree that the revised manuscript is markedly improved and believe the points flagged below should be straightforward to address. Therefore, we invite you to submit a revised version of the manuscript that addresses the points raised during the review process,

We look forward to receiving your revised manuscript.

Kind regards,

Sander L. Hitzig

Academic Editor

PLOS ONE

Journal Requirements:

Reviewers' comments:

Reviewer's Responses to Questions

**Comments to the Author**

1. If the authors have adequately addressed your comments raised in a previous round of review and you feel that this manuscript is now acceptable for publication, you may indicate that here to bypass the “Comments to the Author” section, enter your conflict of interest statement in the “Confidential to Editor” section, and submit your "Accept" recommendation.

Reviewer #1: All comments have been addressed

Reviewer #2: (No Response)

2. Is the manuscript technically sound, and do the data support the conclusions?

Reviewer #1: Yes

Reviewer #2: Yes

3. Has the statistical analysis been performed appropriately and rigorously? 

Reviewer #1: N/A

Reviewer #2: N/A

4. Have the authors made all data underlying the findings in their manuscript fully available?

Reviewer #1: Yes

Reviewer #2: No

5. Is the manuscript presented in an intelligible fashion and written in standard English?

Reviewer #1: Yes

Reviewer #2: Yes

6. Review Comments to the Author

Reviewer #1: This is a dramatically improved manuscript. I have only minor remaining comments.

Figure 1 – The upwards arrows still go beyond the gradually downtrending progression line. Should they just go up to the line? Also, people with PD generally decline after “acute episodes” (sometimes then improving again, but often not back to their prior baseline). The figure might be more helpful if it captures this common response in PD to episodes like acute illness or hospitalization. This common experience is also reflected in the quotes (line 380, for example) so it would be good if the figure reflected this, as well.

Minor – Line 506, missing “w” in “was”

The discussion is markedly improved.

Minor – The sentence of the first paragraph on page 36 of the full PDF (lines 546-550) is difficult to follow. Please break into two sentences and reword for clarity.

Minor – Box 1: There is an empty box under the first subtheme.

With regard to data availabililty, I defer to PLoS One. I agree with the authors that typically transcripts are not submitted for reasons of confidentiality. Sometimes the code book with all relevant quotes is submitted, though.

Reviewer #2: I think the authors have done an excellent job in responding to the reviewers’ comments. In particular, the description of the method is clear and now feels in line with what was carried out and the discussion feels much more thorough with more extensive references to the wider literature.

I have only these very minor comments remaining:

Page 7 (line 169) “lens transitions” I think there is word missing

Page 25 (line 645) “Those with Parkinson’s, and those closest to them, reframe acceptance over the trajectory of the condition” I am wondering if this could be slightly rephrased. As acceptance has not really been covered in the discussion, it seems odd to have it here as the first statement of the conclusion.

Page 26 (Box 1) There is nothing in the strategies box for the subtheme “navigating reactions?” Is something missing from here?

Figure: Thank you for providing the legend which now gives a clear explanation. I’m not sure the thin blue wiggly line adds anything for me (seems to obscure part of the text on my version) but perhaps this looks clearer on other versions. Looking at the blue arrows pointing upwards from the acute episodes box – the final one seems thinner – is that intended (or again, might just be how the formatting turns out on my screen).

7. PLOS authors have the option to publish the peer review history of their article (what does this mean?). If published, this will include your full peer review and any attached files.

Reviewer #1: No

Reviewer #2: No

---

## [Author Response · Author response to Decision Letter 1]

27 Apr 2022

Thank you for your positive review of the manuscript ‘Transitions and challenges for people with Parkinson’s and their family members: A qualitative study’ and for drawing our attention to the minor points raised. A marked-up copy of the manuscript addressing the final minor amendments has been uploaded, and are set out below. We have also uploaded a clean version of the manuscript. 

Reviewer #1: 

1., Figure 1 – The upwards arrows still go beyond the gradually downtrending progression line. Should they just go up to the line? Also, people with PD generally decline after “acute episodes” (sometimes then improving again, but often not back to their prior baseline). The figure might be more helpful if it captures this common response in PD to episodes like acute illness or hospitalization. This common experience is also reflected in the quotes (line 380, for example) so it would be good if the figure reflected this, as well.

The upward arrows have been changed as suggested and the reflection of acute episodes and not returning back to prior baseline.

2., Minor – Line 506, missing “w” in “was”

Line 506 reads “The first point of transition was identified as when participants were told they have a chronic” which is therefore correct.

3., Minor – The sentence of the first paragraph on page 36 of the full PDF (lines 546-550) is difficult to follow. Please break into two sentences and reword for clarity.

This sentence has been broken down and reworded as suggested. 

4., Minor – Box 1: There is an empty box under the first subtheme.

This error has been corrected.

5., With regard to data availabililty, I defer to PLoS One. I agree with the authors that typically transcripts are not submitted for reasons of confidentiality. Sometimes the code book with all relevant quotes is submitted, though.

All relevant data is included in the manuscript itself with quotes from participants. 

Reviewer #2:

6., Page 7 (line 169) “lens transitions” I think there is word missing

Thank you for noticing this, the missing word ‘of’ has now been added.

7., Page 25 (line 645) “Those with Parkinson’s, and those closest to them, reframe acceptance over the trajectory of the condition” I am wondering if this could be slightly rephrased. As acceptance has not really been covered in the discussion, it seems odd to have it here as the first statement of the conclusion.

This has now been reworded to closely reflect the content of the discussion. 

8., Page 26 (Box 1) There is nothing in the strategies box for the subtheme “navigating reactions?” Is something missing from here?

The strategies in box 1 apply to both sub-themes which is now clearer following the removal of the erroneous blank row

9., Figure: Thank you for providing the legend which now gives a clear explanation. I’m not sure the thin blue wiggly line adds anything for me (seems to obscure part of the text on my version) but perhaps this looks clearer on other versions. Looking at the blue arrows pointing upwards from the acute episodes box – the final one seems thinner – is that intended (or again, might just be how the formatting turns out on my screen).

As addressed for reviewer 1, the figure has been amended and the line now reflects more clearly the declines that take place after acute episodes, for example. The arrows are now the same width to one another this may be a formatting issue.

---

## [Editor Report · Decision Letter 2]

3 May 2022

Transitions and challenges for people with Parkinson’s and their family members: A qualitative study

PONE-D-21-37562R2

Dear Dr. Davies,

We’re pleased to inform you that your manuscript has been judged scientifically suitable for publication and will be formally accepted for publication once it meets all outstanding technical requirements.

Kind regards,

Sander L. Hitzig

Academic Editor

PLOS ONE

---

## [Editor Report · Acceptance letter]

12 May 2022

PONE-D-21-37562R2 

Transitions and challenges for people with Parkinson’s and their family members: A qualitative study 

Dear Dr. Davies:

I'm pleased to inform you that your manuscript has been deemed suitable for publication in PLOS ONE. Congratulations! Your manuscript is now with our production department. 

Kind regards, 

on behalf of

Dr. Sander L. Hitzig 

Academic Editor

PLOS ONE